# Have LLMs Advanced Enough?
## A Challenging Problem Solving Benchmark For Large Language Models

**Daman Arora**[*,†]
Microsoft Research
daman1209arora@gmail.com

**Himanshu Gaurav Singh**[*,†]
UC Berkeley
himanshu_singh@berkeley.edu

**Mausam**
IIT Delhi
mausam@cse.iitd.ac.in

## Abstract

The performance of large language models (LLMs) on existing reasoning benchmarks has significantly improved over the past years. In response, we present JEEBENCH, a considerably more challenging benchmark dataset for evaluating the problem solving abilities of LLMs. We curate 515 challenging pre-engineering mathematics, physics and chemistry problems from the highly competitive IIT JEE-Advanced exam. Long-horizon reasoning on top of deep in-domain knowledge is essential for solving problems in this benchmark. Our evaluation on various open-source and proprietary models reveals that the highest performance, even after using techniques like self-consistency, self-refinement and chain-of-thought prompting, is less than 40%. The typical failure modes of GPT-4, the best model, are errors in algebraic manipulation, difficulty in grounding abstract concepts into mathematical equations accurately and failure in retrieving relevant domain-specific concepts. We also observe that by mere prompting, GPT-4 is unable to assess risk introduced by negative marking for incorrect answers. For this, we develop a post-hoc confidence-thresholding method over self-consistency, which enables effective response selection. We hope that our challenging benchmark will guide future re-search in problem-solving using LLMs.

## 1  Introduction

The capabilities of large language models (LLMs) have been improving since the last decade on a plethora of tasks including reasoning. Most recently, GPT-4 demonstrates significant improvements over GPT-3 on tasks such as code-generation, arithmetic and commonsense reasoning (Bubeck et al., 2023), exhibiting impressive performance on standard reasoning and STEM benchmarks such as GSM-8K (Cobbe et al., 2021), MATH (Hendrycks

---

* equal contribution, † work done while at IIT Delhi

A uniform wooden stick of mass 1.6 kg and length $l$ rests in an inclined manner on a smooth, vertical wall of height $h(<l)$ such that a small portion of the stick extends beyond the wall. The reaction force of the wall on the stick is perpendicular to the stick. The stick makes an angle of $30°$ with the wall and the bottom of the stick is on a rough floor. The reaction of the wall on the stick is equal in magnitude to the reaction of the floor on the stick. The ratio $h/l$ and the frictional force $f$ at the bottom of the stick are $(g = 10 \text{ ms}^2)$

(A) $\frac{h}{l} = \frac{\sqrt{3}}{16}, f = \frac{16\sqrt{3}}{3}$ N
(B) $\frac{h}{l} = \frac{3}{16}, f = \frac{16\sqrt{3}}{3}$ N
(C) $\frac{h}{l} = \frac{3\sqrt{3}}{16}, f = \frac{8\sqrt{3}}{3}$ N
(D) $\frac{h}{l} = \frac{3\sqrt{3}}{16}, f = \frac{16\sqrt{3}}{3}$ N

**Solution** Since the stick is in static equilibrium,

1. All forces along the horizontal axis sum to zero.
2. All forces along the vertical axis sum to zero.     **CONCEPT RETRIEVAL**
3. The torque around any point on the stick is zero.

Let the normal reaction at the point of contact between the stick and the wall be $R_1$. $R_1$ will act perpendicular to the stick. Let the normal reaction at the point of contact between the stick and the floor be $R_2$. $R_2$ acts perpendicular to the floor in the upward direction. Let the friction be $f$, acting parallel to the floor.
From (1), we have     **CONCEPT GROUNDING**
$$f = R_1 \cos 30° \quad (1)$$
For applying (3), we use the point of contact between the stick and the floor. Since the torque along it is zero, we have $l_2 = mg$ (2)
$$R_1 \cdot \frac{h}{\cos 30°} = mg \cdot \frac{l}{2} \sin 30° \quad (3)$$
Finally, we are given that
$$R_1 = R_2 \quad (4)$$

Solving equation (2) and (4), we get $R_1 = \frac{mg}{1+\sin 30°} = \frac{2}{3}mg$ Substituting into equation (3), we get     **ALGEBRAIC MANIPULATION**
$$\frac{2}{3}mg\frac{h}{\cos 30°} = mg \cdot \frac{l}{2} \sin 30°$$
$$\frac{h}{l} = \frac{3\cos 30° \sin 30°}{2 \cdot 2} = \frac{3 \cdot \frac{\sqrt{3}}{2} \cdot \frac{1}{2}}{4} = \frac{3\sqrt{3}}{16}$$
From (1), $f = R_1 \cos 30° = \frac{2}{3}mg \cos 30° = \frac{mg\sqrt{3}}{3} = \frac{16\sqrt{3}}{3}$. Therefore, option D is correct.

Figure 1: An example problem from JEEBENCH

et al., 2021b), MMLU (Hendrycks et al., 2021a) and ScienceQA (Lu et al., 2022)

Rising capabilities of LLMs call for harder benchmarks. We introduce JEEBENCH, a benchmark consisting of 515 problems that require complex logical and mathematical reasoning on top of deep in-domain knowledge of pre-engineering level Physics, Chemistry and Mathematics. Problems have been curated from the past 8 editions of the Joint Entrance Examination (JEE)-Advanced exam, held annually in India as an entrance test to India's premier engineering institutes: the IITs. The exam is designed to be time-consuming, diffi-

cult, and has a low selection rate (approx. 5%).

The problems in the dataset require a complex interplay of employing multiple high-level domain specific concepts, grounding them into mathematical equations/constraints, followed by algebraic manipulation and arithmetic operations. Figure 1 is a problem from the dataset along with an expert's solution. In this problem, the ideal solution involves the retrieval of the appropriate concepts: *the rules of static equilibrium*, grounding the concepts into mathematical equations for the specific problem instance, followed by solving the equations in order to find the final answer. Other instances of domain-specific concepts can be *Balancing of redox reactions* (Chemistry), *Current into a junction equals current out of the junction* (Physics) and *Integration by parts* (Mathematics). More such examples can be found in the Appendix A.2.

We conduct a qualitative and quantitative study of contemporary open-source and proprietary LLMs on these problems and also highlight avenues for further research. Our analysis indicates that GPT-4 is unparalleled in performance compared to other models. It demonstrates long horizon reasoning and the ability to manipulate complex algebraic equations in quite a few problems. We observe that chain-of-thought prompting (Kojima et al., 2022) and self-consistency (Wang et al., 2023b), which are recent proposals to improve LLM performance, are indeed effective on our dataset.

We also explore Self-Critique (Madaan et al., 2023; Shinn et al., 2023), where an LLM (verifier) is instructed to improve the outputs of the same LLM (generator). We find that this approach is not helpful on JEEBENCH. The verifier is weak in spotting conceptual errors, and like the generator, is itself prone to hallucinations. It would be interesting to explore the class of problems where this approach of self-refinement is (not) helpful.

We further conduct a critical analysis of the limits of GPT-4's reasoning abilities, and highlight major areas that require considerable improvement. A detailed error analysis suggests that it frequently struggles in retrieving relevant concepts required to solve problems, and performing algebraic manipulation & arithmetic. Inability to perform even simple algebra highlights an important question: can we build LLMs faithful to mathematical logic?

Another important question is how to estimate GPT-4's performance in comparison to humans.

The JEE Advanced Exam comes with the bane of negative marking for incorrectly answered questions. This makes the exam even more challenging, because in addition to advanced problem solving skills, it requires an accurate risk assessment and computing a good policy based on it. Our experiments demonstrate that when prompted with the marking scheme, GPT-4's performance actually drops. To mitigate this, we employ a simple method - *thresholding over self consistency*. Self consistency generates multiple responses for each question. Relative frequency in the set of responses can be considered as a proxy for confidence score of each option. Threshold on the confidence score can be tuned using a validation set. We find that GPT-4's score, after augmenting it this way, lies in the top 10-20 percentile of human scores in the 2023 edition of the exam.

Overall, we hope that this benchmark serves as a strong and reliable test-bed and fosters future research on problem solving with LLMs. Our code and dataset are available at https://github.com/dair-iitd/jeebench.

## 2 Related Work

Reasoning has been studied under various contexts such as logical reasoning, commonsense reasoning, mathematical reasoning, and theorem proving. We summarize some key works in two sub-areas, most closely related to our work: mathematical reasoning and Science QA.

**Mathematical problem solving:** GSM8K (Cobbe et al., 2021), Dolphin18K (Huang et al., 2016), AQuA-RAT (Ling et al., 2017), MATH (Hendrycks et al., 2021b) and Ape210K (Zhao et al., 2020) are datasets that contain mathematical reasoning questions. Dolphin18K, GSM8K, and AQuA-RAT consist of elementary problems, requiring only basic arithmetic and problem comprehension. Thus, there is a general lack of variety in the underlying reasoning steps across problems. In terms of difficulty, MATH, containing problems from AMC, AIME and Olympiads, comes close to JEEBENCH in terms of complexity. However, compared to MATH, the mathematics questions in our dataset span many additional topics such as Differential and Integral Calculus, Differential Equations, 3D geometry, and Conic Sections. Also, the problems in JEEBENCH are harder, as we discuss later in the paper. miniF2F (Zheng et al., 2022) consists of mathematics problems from MATH dataset and

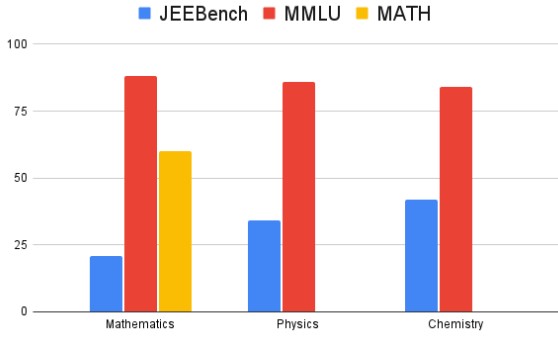

Figure 2: Performance of GPT-4 on a random subset of MATH, MMLU-Physics, Chemistry, Math, JEEBENCH

other sources in a formal language. In contrast, problems in our dataset are in natural language.

**General Science:** In the context of Physics and Chemistry, ScienceQA (Lu et al., 2022), SciQ (Welbl et al., 2017) and MMLU (Hendrycks et al., 2021a) are prominent available datasets. ScienceQA and SciQ, built from elementary and high school science curricula, mainly test factual knowledge of the subject. The skills required to solve such problems are primarily information extraction, reading comprehension and commonsense reasoning. In contrast, questions in our dataset require long-horizon reasoning and grounding of complex scientific concepts into equations and arithmetic. Concurrently, C-Eval (Huang et al., 2023) and SciBench (Wang et al., 2023a) are datasets along similar lines. C-Eval consists of a variety of disciplines such as engineering, medicine and humanities has been created. SciBench creates a dataset from college-level Mathematics, Physics and Chemistry questions.

Problems present in JEEBENCH are significantly harder than those in other contemporary datasets. To verify this, we sample 50 questions each from JEEBENCH and the test sets of MATH and the high-school Physics, Chemistry and Mathematics sections from MMLU and conduct zero-shot evaluations on GPT-4. The results can be seen in Figure 2. As we can see, GPT-4 can easily solve more than 80% problems from MMLU. The MATH dataset is harder, where the performance is approximately 60%. However, GPT-4 struggles in JEEBENCH-Math, solving close to a mere 20% problems.

## 3 The JEEBENCH Dataset

The dataset consists of 515 problems extracted from the past 8 editions of the JEE-Advanced from the year 2016 to 2023. The problems are harvested from publicly available sources.[1] The exam consists of 2 papers held every year, each containing 50-60 questions equally distributed among Physics, Chemistry, and Mathematics. We use online tools to extract problems from PDF-format exam papers into LATEX. We remove all problems containing diagrams in their description (approximately 40%). Manual quality checks are performed to fix/eliminate possible errors in pre-processing. Figure 3 shows representative problems from the final dataset. The problems are categorised by subject: Physics, Chemistry and Mathematics, and the format of expected response: multiple choice questions (MCQ) with single option correct, MCQs with multiple options correct, Integer-type and Numeric-type. In Integer-type questions, the answer is an unbounded non-negative integer, whereas for Numeric-type, the answer is a floating point number upto 2 digits after the decimal point. The breakdown of the problems based on answer-type and subject is shown in Table 1.[2]

|  | **Math** | **Phys** | **Chem** | |
|---|---|---|---|---|
| **Single-Correct** | 53 | 27 | 30 | 110 |
| **Multi-Correct** | 85 | 41 | 60 | 186 |
| **Integer** | 37 | 22 | 23 | 82 |
| **Numeric** | 61 | 33 | 43 | 137 |
| **Total** | 236 | 123 | 156 | 515 |

Table 1: # of questions for each subject and problem-type.

The questions contained in the dataset belong to diverse sub-topics (for example, Math questions could belong to Calculus, Algebra, Combinatorics, etc.). The breakdown of the entire dataset into sub-topics can be found in Appendix A.1.

## 4 Experimental Setup and Results

We wish to investigate the following research problems:

1. How well do LLMs perform on JEEBENCH?

2. How effective are methods, such as chain-of-thought prompting and self-consistency, which have been proposed to improve the reasoning abilities of LLMs?

3. What are the main sources of errors which limit the performance of these models?

[1] https://jeeadv.ac.in/archive.html
[2] There are fewer problems in Physics and Chemistry because more problems in these two subjects contained images.

A gas is enclosed in a cylinder with a movable frictionless piston. Its initial thermodynamic state at pressure $P_i = 10^5$ Pa and volume $V_i = 10^{-3}$ m³ changes to a final state at $P_f = (1/32) \times 10^5$ Pa and $V_f = 8 \times 10^{-3}$ m³ in an adiabatic quasi-static process, such that $P^3V^5 = $ constant. Consider another thermodynamic process that brings the system from the same initial state to the same final state in two steps: an isobaric expansion at $P_i$ followed by an isochoric (isovolumetric) process at volume $V_f$. The amount of heat supplied to the system in the two-step process is approximately
(A) 112 J
(B) 294 J
(C) 588 J
(D) 813 J

What is the total number of compounds having at least one bridging oxo group among the molecules given below?
$N_2O_3$, $N_2O_5$, $P_4O_6$, $P_4O_7$, $H_4P_2O_5$, $H_5P_3O_{10}$, $H_2$ $S_2O_3$, $H_2$ $S_2O_5$

Let $z = \frac{-1+\sqrt{3}i}{2}$, where $i = \sqrt{-1}$, and $r, s \in \{1, 2, 3\}$. Let $P = \begin{bmatrix} (-z)^r & z^{2s} \\ z^{2s} & z^r \end{bmatrix}$ and $I$ be the identity matrix of order 2 . Then what is the total number of ordered pairs $(r, s)$ for which $P^2 = -I$?

Figure 3: Instances from the dataset from each subject: Physics (Top), Chemistry (Middle), Math (Bottom)

4. Can LLMs be used to verify their own generations in the context of JEEBENCH? What are the limitations in this behaviour?
5. How would they perform in an exam setting, where each question could potentially give negative marks when answered incorrectly?

## 4.1 Metrics

For Single-Correct MCQs and Integer-type questions, we use accuracy as the metric, that is, a score of 1 if the model response matches the gold response, otherwise 0. For Numeric-type questions, we award a score of 1 if the model response differs from the gold response by atmost 0.01. For Multi-Correct MCQs, we award a score of 1 if the model response matches all the correct options. If any of the options selected by the model is incorrect, we award 0. If the model selects some of the correct options and no incorrect option, then for each correct option in the output, the model is given a score of 0.25. For example, if the gold response is ABD and the output response is BD, a score of 0.5 is awarded. This is done, so as to reflect the actual scoring method of JEE-Advanced, which incentivizes a student to not guess.[3]

## 4.2 Prompting LLMs

We evaluate the proposed benchmark on some open-source models: Falcon7B-Instruct (Almazrouei et al., 2023) and Alpaca-LoRA, which uses low-rank adapation (Hu et al., 2021) to reproduce Alpaca (Taori et al., 2023). Then, we evaluate proprietary models such as the OpenAI's

---

[3]JEE Advanced also employs negative marking. We incorporate it in Section 4.5 while comparing with human performance in exam setting.

GPT series of models `text-davinci-003` (GPT-3), `gpt-3.5-turbo` (GPT-3.5) and `gpt-4-0314` (GPT-4) , as well as `text-bison-001` (PaLM-2) provided by Google. Evaluation on larger open-sourced LLMs is left for future work.

For obtaining the model's response, each model is prompted with the expected response type concatenated with the problem description. The exact system and user prompts can be found in the Appendix A.3. The exact answer is extracted manually from the response generated by the LLM. Sometimes, the LLM response is gibberish and sometimes responds by saying that none of the options are correct. For both of these cases, we record "None" as the answer. If the question's expected response type doesn't match the question type (for example non-integer for an integer type question), even then we record it as a "None" response.

We also conduct a few-shot evaluation with examples drawn from 2014 edition of the exam. One example is chosen for each question type and subject pair.

All the proprietary models were prompted between May 17, 2023 and June 23, 2023. The maximum response length is set to 2048 and decoding temperature is set to 0. Table 2 contains the results obtained on various LLMs aggregated by subject and question type.

**General trends:** We observe that open-source models perform as good as random and are, in general, lagging behind proprietary models. Performance on JEEBENCH increases consistently with newer versions of the GPT model. GPT-3 exhibits near random performance, but GPT-3.5 and GPT-4 perform significantly better. GPT-4 is far superior to GPT-3.5, by a large margin of 12.9 points but overall performance still remains close to 30%. It is evident that the performance boost is the highest for Chemistry, followed by Physics, and lastly Maths. This is probably because the complexity of reasoning is highest in Mathematics questions and least in Chemistry in JEEBENCH. These results highlight the difficulty of the benchmark posed to both open-source and proprietary models.

Hereafter, we focus on just GPT-4's performance since it is far superior to other models. Firstly, we evaluate the performance of methods like zero-shot chain-of-thought prompting (Kojima et al., 2022), self-consistency (Wang et al., 2023b) & self-refinement (Madaan et al., 2023) on JEEBENCH.

**Zero shot Chain-of-Thought prompting:** The

| | Chemistry | Mathematics | Physics | Integer | Single-Correct | Multi-Correct | Numeric | Total |
|---|---|---|---|---|---|---|---|---|
| **Random** | 0.108 | 0.105 | 0.103 | 0.000 | 0.250 | 0.144 | 0.000 | 0.105 |
| **Alpaca-LoRA** | 0.072 | 0.101 | 0.087 | 0.037 | 0.164 | 0.122 | 0.015 | 0.089 |
| **Falcon7B-Instruct** | 0.083 | 0.114 | 0.085 | 0.000 | 0.182 | 0.142 | 0.029 | 0.098 |
| **GPT-3** | 0.135 | 0.107 | 0.134 | 0.049 | 0.291 | 0.133 | 0.015 | 0.122 |
| **PaLM2** | 0.192 | 0.130 | 0.146 | 0.073 | 0.291 | 0.165 | 0.073 | 0.153 |
| **GPT-3.5** | 0.228 | 0.146 | 0.173 | 0.073 | 0.318 | 0.249 | 0.029 | 0.177 |
| **GPT-4** | 0.423 | 0.212 | 0.352 | 0.207 | 0.455 | 0.383 | 0.153 | 0.309 |
| **GPT-4+CoT** | 0.468 | 0.280 | 0.335 | 0.256 | 0.473 | 0.448 | 0.175 | 0.350 |
| **GPT-4+ (1-shot) CoT** | 0.409 | 0.198 | 0.323 | 0.244 | 0.391 | 0.340 | 0.175 | 0.292 |
| **GPT-4+CoT+Self Critique** | **0.487** | 0.234 | 0.352 | 0.280 | 0.355 | **0.444** | 0.219 | 0.339 |
| **GPT-4+CoT+SC@8** | 0.463 | **0.308** | **0.449** | **0.293** | **0.618** | 0.410 | **0.234** | **0.389** |

Table 2: This table shows the score obtained by various open-source and proprietary models on JEEBENCH aggregated by subject on the left question type on the right. The overall aggregate scores are in the last column. Note that CoT in the table refers to zero-shot CoT except for GPT-4 + (1-shot) CoT.

original prompt is concatenated with the phrase *Let's think step by step*, as proposed by Kojima et al. (2022). We observe that this approach leads to significant improvement in performance, improving vanilla GPT-4 by 4.2 points.

**Few-shot Chain-of-Thought prompting:** We prepend the question with one few-shot example for each question-type, subject pair. Overall, 1-Shot CoT achieves a score of 0.296 as opposed to zero-shot CoT at 0.350 and vanilla GPT-4 at 0.308. Our hypothesis is that few-shot prompting is not very helpful in these questions, because conceptual errors are hard improve upon using few-shot examples. Additionally, many novel reasoning paths might not be covered in the few-shot examples. Thus, our dataset acts as an interesting testbed for advanced approaches in few-shot prompting. Similar results where scores are better with zero-shot CoT as compared to few-shot CoT have been found in Wang et al. (2023a).

**Function calling:** Since GPT-4 makes a lot of arithmetic errors, we decide to also test the function-calling API exposed by OpenAI. Since the JEE exam only allows access to a basic calculator with 4 primitive arithmetic operation(+, -, /, *), allowing plugins such as Wolfram or python would not make a fair comparison (for example, directly asking Wolfram to integrate a function instead of doing it from first principles). Instead, to ensure a level-playing field, we use the newly introduced function-calling API to implement standard arithmetic operators.

Note that function-calling is only allowed with gpt-4-0613 whereas the rest of our results are with gpt-4-0314. We first evaluate gpt-4-0613 with zero-shot CoT. Surprisingly, results suggest that gpt-4-0613 (the new version) is weaker with a CoT

Figure 4: The figure shows the different types of error made by GPT-4 in its response. (i) (top) exhibits a computation error, where the squaring operation performed is algebraically wrong. (ii) (middle) response shows a conceptual error where the model is unable to retrieve the relevant concepts required to solve the problem (iii) (bottom) response is a grounding error, where the concept is correct, however the application in terms of computing # lone pair electrons on Br in $BrF_5$ is wrong.

performance of 0.303 as compared to 0.350 with gpt-4-0314.

Using a calculator API reduces performance even more to 0.274 from 0.303. We observe that tool usage isn't very robust for GPT-4, where it sometimes hallucinates invalid function arguments and sometimes even invalid function names! Also, GPT-4 is quite accurate at arithmetic for small digit operations. Computation errors are mostly during symbolic manipulation, rather than purely arithmetic operations, which is probably why a black-box calculator isn't very beneficial.

**Self-Consistency (SC):** We sample multiple responses from the LLM at a non-zero temperature.

For Integer-type, Numeric-type and Single-Correct MCQs, we use a majority vote (from all the responses which are not "None") as the proposed answer. For Multi-Correct MCQs, we choose a simplifying assumption that all options are independent. If an option occurs atleast 50% times in the responses, we select it, otherwise we don't. We use $\tau = 0.5$ and the number of responses is set to 8. Self-consistency helps a lot in improving over the GPT-4+CoT baseline by a score of +3.9 points. In the future, it will be interesting to apply extensions such as adaptive consistency for better cost-quality tradeoffs (Aggarwal et al., 2023).

### 4.3 Error Analysis of System Responses

In order to assess GPT-4's weaknesses, we conduct a manual inspection of the errors it makes in its reasoning chains. We perform this study on the errors made by GPT-4+CoT on a random subset of 100 problems. The score obtained on this subset is 27.25. We ask the following questions about the model response for each problem instance:

1. Is GPT-4 able to retrieve the concepts/facts required for solving the problem? Inability to do this contributes to *conceptual* errors.
2. If relevant concepts are retrieved, are they grounded correctly as equations/constraints? These contribute to *grounding* errors.
3. Is the algebraic manipulation & arithmetic correct? These contribute to *computation* errors.

Refer to Figure 4 for an illustration of each type of error[4]. In one case, we find that GPT-4 misunderstands the question. The overall results of this analysis is shown in Table 3.

| Error Type | Count |
|---|---|
| Conceptual Error | 34 |
| Computation Error | 30 |
| Grounding Error | 15 |
| Problem Miscomprehension | 1 |
| Perfect | 20 |

Table 3: Variety of errors GPT-4 makes in the solution.

Our error analysis indicates that most errors are caused because of not being able to retrieve concepts (34 out of 80), which are critical to making progress in the solution, or due to computation errors (30 out of 80). Moreover, in 20 questions,

---

[4]Note that a question can have multiple types of errors, however, we only investigate the first error that is noticed.

where the answer is correct (out of 27), the explanation is also correct. i.e., 28% of the time, the model gives a correct answer for the wrong reasons.

### 4.4 Can GPT-4 find and correct its mistakes?

Can GPT-4 be used to grade its own outputs? A good grader should be able to spot errors in a solution. Using an LLM to critique its own output has been proposed recently by multiple works (Shinn et al., 2023; Madaan et al., 2023) and has shown improvements on some datasets. A good verifier should be able to catch and fix all errors. Even when the final answer is correct, it isn't necessary that intermediate reasoning steps are correct.

We put the idea of self-critique to test on JEEBENCH. After a CoT response has been generated, we prompt another GPT-4 instance by first describing the problem, GPT's solution and then appending the instruction: *"Find problems(if any) with the given solutions. If there are any errors, correct it and give the new answer."*

We re-evaluate the new answer suggested by GPT-4. Results clearly show that this approach doesn't lead to improvement. In fact, it leads to poorer results as compared to GPT-4+CoT and the performance goes down from 35% to 33.9%.

In order to develop a deeper understanding of the repairs suggested by the verifier GPT-4, a manual inspection is performed. We use the same subset of 100 problems picked up earlier for categorizing error-types. For each generated solution and suggested edit, we pose the following questions:

- Can the verifier *find* problems in the solution?
- Can the verifier *fix* problems if it finds them?
- Is the problem identified by the verifier actually a valid problem?

| Error in solution? | Verifier Response | Count |
|---|---|---|
| Yes | No error found | 46 |
| | Found error but didn't fix it | 25 |
| | Converted non-error to error | 7 |
| | Found error and fixed it | 2 |
| No | Converted non-error to error | 1 |
| | Didn't find error | 19 |

Table 4: The figure shows the breakup of the kind of errors the verifier GPT-4 makes while suggesting edits.

Our results can be seen in Table 4. It is evident that, contrary to observations in other works, on JEEBENCH, GPT-4 is mostly ($\frac{46}{80} = 57.5\%$) unable to find errors in solutions it proposes. Even when it can, it is unable to fix them. Only in 2

**Question**
Let $S$ be the set of all complex numbers $z$ satisfying $|z^2 + z + 1| = 1$. Then which of the following statements is/are TRUE?
  (A) $|z + \frac{1}{2}| \leq \frac{1}{2}$ for all $z \in S$
  (B) $|z| \leq 2$ for all $z \in S$
  (C) $|z + \frac{1}{2}| \geq \frac{1}{2}$ for all $z \in S$
  (D) The set $S$ has exactly four elements
**Answer**
First, we can rewrite the given equation as follows:
$|z^2 + z + 1| = 1 \implies |z^2 + z + 1 - 1| = 0 \implies |z^2 + z| = 0$

---

There is an error in the given solution. The equation $|z^2 + z| = 0$ was incorrectly derived from the original equation $|z^2 + z + 1| = 1$. The correct derivation should be:
$|z^2 + z + 1| = 1 \implies |z^2 + z + 1 - 1| = 1 \implies |z^2 + z| = 1$

---

**Question**
A metal is heated in a furnace where a sensor is kept above the metal surface to read the power radiated $(P)$ by the metal. The sensor has a scale that displays $\log_2 (P/P_0)$, where $P_0$ is a constant. When the metal surface is at a temperature of 487°C, the sensor shows a value 1. Assume that the emissivity of the metallic surface remains constant. What is the value displayed by the sensor when the temperature of the metal surface is raised to 2767°C ?

Step 5: Calculate the final answer. Now, we just need to plug in the values of $T_1$ and $T_2$ and calculate the final answer: $\log_2 \left(2 \cdot \frac{(3040.15)^4}{(760.15)^4}\right) = \log_2(2 \cdot 16^4) =$

The given solution is correct, and the final answer is 17. The steps are well-explained, and the calculations are accurate.

Figure 5: [Top]: Question where GPT-4 identifies a mistake but is unable to fix it. The problem and part of the response is on the top. The bottom block contains the edit suggested by GPT-4. The manipulation in the edit suggested is mathematically wrong. [Bottom]: Question where GPT-4 is unable to identify an error. The problem and part of the response is on the top. The bottom block contains the edit suggested by GPT-4. It should be $log_2(2 \cdot 4^4)$ instead of $log_2(2 \cdot 16^4)$
.

out of 80 questions, was GPT-4 able to give a meaningful edit to an erroneous solution, which is over-compensated by the solutions it degrades by suggesting edits to parts of the solution which are already correct. Figures 5 provides example of errors made by the verifier. The complete response for these along with other examples can be found in the Appendix A.5. This experiment raises an interesting question: for what class of problems is self-critique (not) helpful? It might be interesting to look into methods which use learnt verifiers (Arora and Kambhampati, 2023; Lightman et al., 2023).

## 4.5 Comparison with human performance

The JEE exam contains negative marking, for instance, single-correct MCQ questions are awarded a score of +3 marks when correct, -1 if answered incorrectly and 0 when not answered. For Mult-Correct MCQs, a score of +4 is given when all options are included in the final answer. If any of the options is wrong, -2 is given. If some of the options are correct, +1 is given for each correct option. The skills needed for an examinee to maximize one's score include being able to assess one's own confidence in their response and being able to decide whether to answer or not based on the con-

fidence levels. Contingent on the former skill, the latter is a simple decision-theoretic computation under uncertainty.

### 4.5.1 Deciding whether to answer

To attain a good score in the examination, it is important to ensure that the model does not answer when it is unsure of its solution. Can LLMs assess this risk and plan accordingly when prompted with the marking scheme? To investigate this, we prompt the model with the exact marking scheme for each MCQ question type along with problem statement, and then ask to generate an answer or skip the question altogether. The complete prompt is in Appendix A.6. We re-run inference on all the problems with these prompts for all the MCQ questions. The results can be seen in Table 5.

| Method | Pos. Score | Neg. Score | Total |
|---|---|---|---|
| GPT-4+CoT w/o Marking | 489 | 181 | 308 |
| GPT-4+CoT w Marking | 404 | 206 | 198 |

Table 5: Marks obtained when GPT-4 is prompted with the marking scheme v/s without on MCQ questions. These marks are out of a total of 1074.

Results indicate that prompting is not helpful in this case, and GPT-4 cannot effectively decide when not to answer. This is in line with observations made by Valmeekam et al. (2022) where it is shown that LLMs have poor planning capabilities. In response, we develop a post-hoc confidence-thresholding method on self-consistency responses.

### 4.5.2 Calibration

For single-correct & multiple-correct MCQs, we compute the confidence score for each option by computing its relative frequency in the set of responses. Note that often, GPT-4 is unable to answer the question at all, or arrives at a conclusion that is not supported by any option (a "None" response). In such cases, we do not count contributions from this response. For instance, if a model's response in 4 attempts in a Multi-Correct MCQ is "AB", "None", "B", "AC", then, the confidence for options are A:$\frac{1}{2}$, B:$\frac{1}{2}$, C:$\frac{1}{4}$, D:0.

Figure 6 is the calibration cruve of GPT-4 on JEEBENCH. The maximum calibration Error (MCE) is 0.136 and the average calibration error (ACE)[5] is 0.098. The plot suggests that the model is slightly overconfident at high confidences, because of lower accuracy on higher levels of confidence,

---
[5]MCE/ACE are the maximum/average absolute difference b/w confidence & accuracy among all confidence bins.

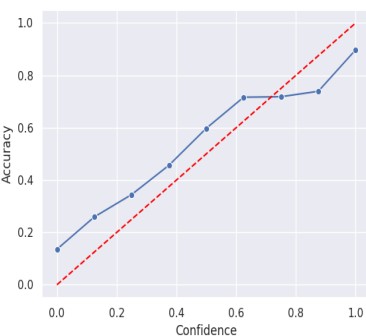

Figure 6: Calibration plot of GPT-4 on MCQ questions

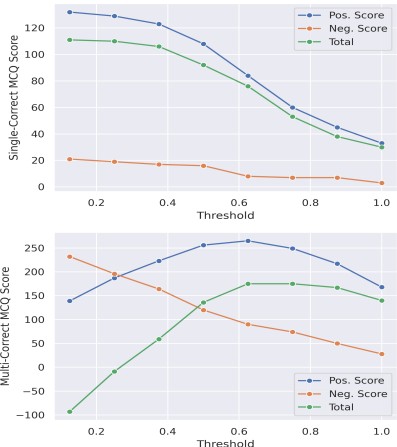

Figure 7: Scores obtained on different thresholding values on Single-Correct(top) and Multi-Correct(bottom) type questions from the val set, the optimal value is $\tau_{single} = 0.125$ and $\tau_{multiple} = 0.75$

but slightly underconfident at low and medium confidences.

### 4.5.3 Thresholding with Self-Consistency

Our objective is to decide whether to include an option in the final response or not. We wish to compute a parameter $\tau$ such that an option will be in the final response if the confidence for that option is atleast $\tau$. We compute separate $\tau_{single}, \tau_{multiple}$ for Single-correct and Multiple-correct MCQs respectively. We compute confidence scores for GPT-4's response to each question as in Section 4.5.2. Questions from 2016-2021 are chosen as the validation set and from 2022-2023 as the test set. The best $\tau_{single}$ and $\tau_{multiple}$ thresholds for Single-Correct and Multi-Correct MCQs by simple hyper-parameter search. Figure 7 shows the plot of positive, negative and total score on the validation set over range of possible values of $\tau_{single}$ and $\tau_{multiple}$. The optimal value of $\tau_{multiple}$ is 0.75 and of $\tau_{single}$ is 0.125. $\tau_{single}$ being less than 0.25 indicates that taking a majority vote is the

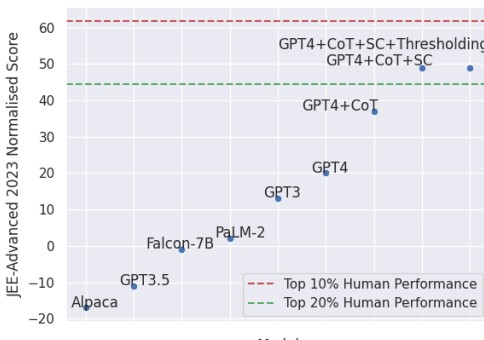

Figure 8: Plot showing performance of models compared to projected human performance.

best strategy for single-correct MCQs. However, this is not true for multi-correct MCQs, where a threshold of $\tau_{multiple} = 0.5$ (as done originally) is sub-optimal. We assume that Integer and Numeric questions do not have negative marking. The final response for them is decided using a majority vote over responses. Table 6 shows scores with the optimal thresholds on the test set. We find that not answering when confidence is less than threshold increases the total score by about 4.3%.

| Method | Pos Score | Neg Score | Total Score |
|---|---|---|---|
| GPT-4+CoT | 109 | 43 | 66 |
| GPT-4+CoT+SC | 118 | 49 | 69 |
| GPT-4+CoT+SC+Thresholding | 111 | 39 | **72** |

Table 6: Marks on the test set obtained when optimal thresholds derived from the val set are used.

### 4.6 Estimating performance compared to humans

Finally, we wish to estimate the performance of GPT-4 compared to humans. For this we use the 2023 examination paper since there is almost no probability of contamination. The 2023 paper was released on 4th June 2023 and contained 65 questions which were textual (rest 37 contained images). The total marks in the exam were 360. The score obtained by GPT-4 after confidence-thresholding on MCQs and regular aggregation on Integer and Numeric types is 49 out of 229. Assuming average difficulty levels in questions that were not included (because they contained images) is equal to ones that were, we normalize the projected human performance from 229 to 360 giving it a total of 77 marks out of 360. Projections indicates that this would place GPT-4 around the 80-90 percentile range. The results of JEE Advanced 2023 indicate that the top 10% score is approximately 97/360 and top 20% score is approximately 70/360. Figure 8

### 4.7 Has GPT-4 memorized some problems?

In the era of internet-scale pre-training, it is very hard to ascertain whether a dataset has been used for training a particular model. Nevertheless, we tried to investigate the contamination of JEEBENCH. This was done by (i) searching for instances in JEEBENCH from publicly available internet corpora, (ii) prompting the LLM to complete the problem statement itself, when prompted with a prefix of the problem statement. Both these investigations suggest only minor (approx. 6%) contamination. A detailed description of our contamination study can be found in Appendix A.8.

GPT-4+CoT+SC attains a score of 0.338 on Advanced 2023 questions, which is not very far from the aggregate performance of 0.396 on the remaining dataset. Given that 2023 questions are uncontaminated, we believe that the extent of contamination is quite low, and its performance on this dataset is a genuine indication of its current reasoning abilities. It is also noteworthy that some exams (e.g., JEE Advanced 2017) are easier, and GPT-4 does much better on it raising the aggregate score.

## 5 Discussion

The general performance trend demonstrates the efficacy of high-quality data, instruction fine-tuning, RLHF and parameter scaling in improving the reasoning capabilities of LLMs. For many problems, GPT-4 is able to give a sketch of the correct, human-like solution that is impressive given the extent of reasoning involved in the problems. However, our analysis also reveals major areas where progress is needed. Although GPT-4 performs flawless logical and mathematical reasoning in some instances, sometimes it commits grave errors in trivial steps.

Errors in retrieval and application of concepts suggest an interesting research question: Can we augment an LLM such that it's generation is constrained by faithfulness to a set of facts? Such a system will demonstrate robustness in reasoning, critical for long-horizon tasks.

Physics problems in the benchmark often require an understanding of spatial reasoning. We found that while GPT-4's spatial reasoning is far from perfect. Appendix A.7 provides an example where GPT-4 commits errors which might be attributed to its inability to reason spatially. With the release of the multi-modal version of GPT-4, evaluating this aspect of Physics problems might be easier.

Finally, an LLM that understands its own confidence in an answer is a key missing piece, as highlighted by our experiments in the exam setting. Our simple post-hoc wrapper does slightly improve performance in this regard.

## 6 Conclusion

We present a challenging problem solving benchmark to evaluate large language models. We perform a detailed analysis of the performance of various LLMs on the benchmark, and identify areas of improvement in the best current LLMs. Our work raises interesting research directions such as mathematical logic-augmented GPT, multi-modal evaluations on GPT-4 and the decision-making capabilites of GPT in an exam setting. We hope that JEEBENCH guides future research in reasoning using LLMs. Our code and dataset are available at https://github.com/dair-iitd/jeebench.

### Acknowledgements

We thank Dr. Parag Singla and the JEE Office for helping with getting approvals to use the dataset for research purposes. We thank Rishabh Ranjan for discussions around calibration and Mohd. Zaki for help in obtaining OpenAI API access for this work. The work is supported by grants from Google, Verisk, Microsoft, and Huawei, and the Jai Gupta chair fellowship by IIT Delhi. We thank the IIT-D HPC facility for its computational resources.

### Limitations

Contamination is a big problem in the era of pre-trained language models which have been trained on large web corpora. Therefore, its really hard to determine if a dataset has been seen. Determining the extent of contamination is also not easy, although we make an effort to quantify it. Evaluations against humans is also a slightly flawed process due to other limitations such as time pressure during the examination procedure. Additionally, this data's distribution is fixed to pre-college Physics, Chemistry and Mathematics. There are more gradations and difficulty levels at which the model can be evaluated which have not been tested as part of our analysis.

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

## A Appendix

### A.1 Sub-topic wise distribution of questions in JEEBENCH

Figure 9 provides the topic wise distribution of problems in JEEBENCHfor each subject.

### A.2 Example problems from JEEBENCH

Here we present a few problems from JEEBENCHalong with expert written solutions, with concepts being highlighted in yellow, their grounding being highlighted in violet, and the final algebraic manipulation highlighted in green. See Figures 10 for a Math Problem, Figure 11 for a Physics problem and Figure 12 for a Chemistry problem.

### A.3 Exact Prompts for GPT models

For each problem, we prompt the model with the required answer type and then the prompt. For different response types, the prompts are:

1. **Single-Correct:** `In this problem, only one option will be correct. Give a detailed solution and end the solution with the final answer.`

2. **Multi-Correct:** `In this problem, multiple options can be correct. Give a detailed solution and end the solution with the final answer.`

3. **Integer:** `In this problem, the final answer will be a non-negative integer. Give a detailed solution and end the solution with the final answer.`

4. **Numeric:** `In this problem, the final will be a numeric value. Give the numerical answer correct upto the 2nd decimal digit. Give a detailed solution and end the solution with the final answer.`

The entire prompt consists of the question prepended with the response type prompt. An example is:

In this problem, only one option will be correct.

Give a detailed solution and end the solution with the final answer.

Problem: A uniform wooden stick of mass 1.6 kg and length $l$ rests in an inclined manner on a smooth, vertical wall of height $h(<l)$ such that a small portion of the stick extends beyond the wall.

The reaction force of the wall on the stick is perpendicular to the stick. The stick makes an angle of $30°$ with the wall and the bottom of the stick is on a rough floor. The reaction of the wall on the stick is equal in magnitude to the reaction of the floor on the stick. The ratio $h/l$ and the frictional force $f$ at the bottom of the stick are $\left(g = 10 \text{ ms s}^2\right)$

(A) $\frac{h}{l} = \frac{\sqrt{3}}{16}, f = \frac{16\sqrt{3}}{3}$ N

(B) $\frac{h}{l} = \frac{3}{16}, f = \frac{16\sqrt{3}}{3}$ N

(C) $\frac{h}{l} = \frac{3\sqrt{3}}{16}, f = \frac{8\sqrt{3}}{3}$ N

(D) $\frac{h}{l} = \frac{3\sqrt{3}}{16}, f = \frac{16\sqrt{3}}{3}$ N

### A.4 Errors made by GPT-4

Here we present some errors made by GPT-4.

#### A.4.1 Conceptual Error

This is an example of a Conceptual Error

What is the total number of distinct $x \in [0,1]$ for which $\int_0^x \frac{t^2}{1+t^4} dt = 2x - 1$?

**GPT-4 Response**

First, we can differentiate both sides of the equation with respect to $x$ to get rid of the integral. Using the Fundamental Theorem of Calculus, we have:

$\frac{d}{dx} \left( \int_0^x \frac{t^2}{1+t^4} dt \right) = \frac{d}{dx}(2x - 1)$

$\frac{x^2}{1+x^4} = 2$ This is an error because if two functions are equal at a point, that doesn't imply that their derivatives are also equal.

Now, we want to find the number of distinct solutions for $x$ in the interval $[0,1]$. Let's first solve the equation for $x$:

$x^2 = 2(1 + x^4)$

$x^2 - 2x^4 - 2 = 0$

This is a quartic equation, and we can try to find its roots. Notice that $x = 0$ is not a solution, so we can divide both sides by $x^2$:

$1 - 2x^2 - 2x^{-2} = 0$

Now, let $y = x^2$. Then, we have:

$1 - 2y - 2y^{-1} = 0$

$2y^2 - y + 1 = 0$

This is a quadratic equation in $y$. We can use the quadratic formula to find its roots:

$y = \frac{-(-1) \pm \sqrt{(-1)^2 - 4(2)(1)}}{2(2)}$

$y = \frac{1 \pm \sqrt{1-8}}{4}$

Since the discriminant is negative ($1 - 8 = -7$), there are no real roots for $y$. Therefore, there are no real solutions for $x$ in the given equation.

Thus, the total number of distinct $x \in [0,1]$ for which $\int_0^x \frac{t^2}{1+t^4} dt = 2x - 1$ is $\boxed{0}$.

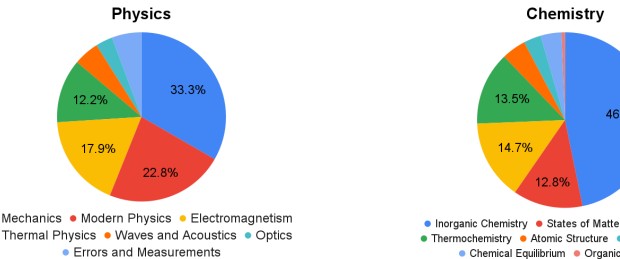
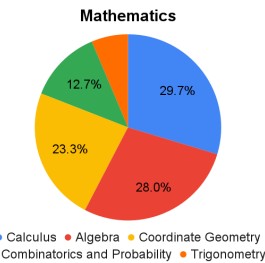

Figure 9: Topic-wise distribution of problems in our dataset

Let $f : (0, \infty) \to R$ be a differentiable function such that $f'(x) = 2 - \frac{f(x)}{x}$ for all $x \in (0, \infty)$ and $f(1) \neq 1$. Then
(A) $\lim_{x \to 0+} f'\left(\frac{1}{x}\right) = 1$
(B) $\lim_{x \to 0+} xf\left(\frac{1}{x}\right) = 2$
(C) $\lim_{x \to 0+} x^2 f'(x) = 0$
(D) $|f(x)| \leq 2$ for all $x \in (0, 2)$

**Solution**

First, we are given that $f'(x) = 2 - \frac{f(x)}{x}$ for all $x \in (0, \infty)$. We can rewrite this as a first-order linear differential equation:

$$f'(x) + \frac{f(x)}{x} = 2.$$

To solve this differential equation, we can use an integrating factor. The integrating factor is given by $e^{\int \frac{1}{x} dx} = e^{\ln x} = x$. Multiplying both sides of the differential equation by the integrating factor, we get:

$$xf'(x) + f(x) = 2x.$$

Now, we can notice that the left-hand side is the derivative of the product $xf(x)$:

$$\frac{d}{dx}(xf(x)) = 2x.$$

Integrating both sides with respect to $x$, we get:

$$xf(x) = x^2 + C,$$

where $C$ is the constant of integration. To find the value of $C$, we can use the given condition $f(1) \neq 1$. Plugging in $x = 1$, we get $f(1) = 1 + C$. Since $f(1) \neq 1$, we must have $C \neq 0$. Now, we can write the function $f(x)$ as $f(x) = x + \frac{C}{x}$, and $f'(x) = 1 - \frac{C}{x^2}$.
For option A, $f'(\frac{1}{x}) = 1 - Cx^2$, therefore $\lim_{x \to 0+} f'(\frac{1}{x}) = 1$.
For option B, $xf(\frac{1}{x}) = x(\frac{1}{x} + Cx) = 1 + Cx^2$, therefore $\lim_{x \to 0+} xf(\frac{1}{x}) = 1 \neq 2$
For option C, $x^2 f'(x) = x^2 - C$, therefore $\lim_{x \to 0+} x^2 f'(x) = -C \neq 0$
For option D, since $C$ can be any arbitrary non-zero value, $f(x)$ is unbounded in $(0, 2)$.
Therefore, the answer is $\boxed{A}$.

Figure 10: A Math problem

A person measures the depth of a well by measuring the time interval between dropping a stone and receiving the sound of impact with the bottom of the well. The error in his measurement of time is $\delta T = 0.01$ seconds and he measures the depth of the well to be $L = 20$ meters. Take the acceleration due to gravity $g = 10 \text{ ms}^{-2}$ and the velocity of sound is $300 \text{ ms}^{-1}$. Then the fractional error in the measurement, $\delta L/L$, is closest to
[A] 0.2%
[B] 1%
[C] 3%
[D] 5%

**Solution**

The time taken to hear the sound of impact would be the time taken for the stone to reach the bottom of the well in addition to the time taken for the sound generated to reach the top of the well.
Since the stone is dropped from rest, we have that $L = \frac{1}{2}gt^2$, therefore $t = \sqrt{\frac{2L}{g}}$. The time taken for the sound to reach the top of the well is $\frac{L}{v}$ where $v$ is the velocity of sound. Therefore,

$$T = \sqrt{\frac{2L}{g}} + \frac{L}{v}$$

Since we need to find the fractional error in the measurement of $L$, that is $\frac{\delta L}{L}$, we can differentiate both side of the equation with respect to $L$. Using this, we get

$$\frac{\delta T}{\delta L} = \frac{1}{2}\sqrt{\frac{2}{Lg}} + \frac{1}{v}$$

$$\delta L = \frac{\delta T}{\frac{1}{2}\sqrt{\frac{2}{Lg}} + \frac{1}{v}}$$

$$\frac{\delta L}{L} = \frac{\delta T}{L(\frac{1}{2}\sqrt{\frac{2}{Lg}} + \frac{1}{v})} = \frac{\delta T}{\sqrt{\frac{L}{2g}} + \frac{L}{v}}$$

$$\frac{\delta L}{L} = \frac{0.01}{\sqrt{\frac{20}{2 \cdot 10}} + \frac{20}{300}} = \frac{0.01}{\frac{16}{15}} = \frac{15}{16}\% \approx 1\%$$

Figure 11: A Physics problem

## A.4.2 Grounding Error

This is an example of a Grounding Error

A plano-convex lens is made of a material of refractive index $n$. When a small object is placed 30 cm away in front of the curved surface of the lens, an image of double the size of the object is produced. Due to reflection from the convex surface of the lens, another faint image is observed at a distance of 10 cm away from the lens. Which of the following statement(s) is(are) true?

(A) The refractive index of the lens is 2.5

(B) The radius of curvature of the convex surface is 45 cm

(C) The faint image is erect and real

(D) The focal length of the lens is 20 cm

**GPT-4 Response**

First, let's consider the refraction of light through the lens. Since the image formed is double the size of the object, we can say that the image distance ($v$) is twice the object distance ($u$). So, $v = 2u = 60$ cm. This is an example of wrong grounding. The equation generated is inconsistent with the sign convention which must be taken care of for the other equations to be valid.

Now, let's use the lens maker's formula to find the focal length ($f$) of the lens: $\frac{1}{f} = (n - 1)\left(\frac{1}{R_1} - \frac{1}{R_2}\right)$

Since the lens is plano-convex, one surface is flat, and its radius of curvature ($R_1$) is infinite. The radius of curvature of the convex surface ($R_2$) is negative because it is convex. So, the formula becomes: $\frac{1}{f} = (n-1)\left(0 - \frac{1}{-R_2}\right)$ $\frac{1}{f} = (n-1)\frac{1}{R_2}$

Galena (an ore) is partially oxidized by passing air through it at high temperature. After some time, the passage of air is stopped, but the heating is continued in a closed furnace such that the contents undergo self-reduction. What is the weight (in kg ) of Pb produced per kg of $O_2$ consumed?
(Atomic weights in $gmol^{-1}$ : $O = 16$, $S = 32$, $Pb = 207$ )

**Solution**

Galena(lead sulfide) when partially oxidized converts to lead oxide which further reacts with the lead sulfide to give lead metal. The reaction is as follows:

$$2PbS + 3O_2 \rightarrow 2PbO + 2SO_2$$

$$PbS + 2PbO \rightarrow 3Pb + SO_2$$

The net reaction can be written as

$$PbS + O_2 \rightarrow Pb + SO_2$$

Therefore, moles of lead metal produced will be equal to moles of oxygen consumed. Therefore, we have

$$\frac{\text{weight of oxygen}}{\text{molar mass of oxygen}} = \frac{\text{weight of lead}}{\text{molar mass of lead}}$$

$$\frac{1}{32} = \frac{\text{weight of lead}}{207}$$

Therefore, weight of lead produced per kg of oxygen is $207/32 = 6.47$ kg

Figure 12: A Chemistry problem

Now, let's use the lens formula to relate $u$, $v$, and $f$: $\frac{1}{f} = \frac{1}{u} + \frac{1}{v}$

Substituting the values of $u$ and $v$: $\frac{1}{f} = \frac{1}{30} + \frac{1}{60} = \frac{3}{60} = \frac{1}{20}$

Now, we can substitute the expression for $\frac{1}{f}$ from the lens maker's formula: $(n-1)\frac{1}{R_2} = \frac{1}{20}$

Now, let's consider the reflection of light from the convex surface of the lens. Since the faint image is formed at a distance of $10$ cm from the lens, we can use the mirror formula to relate the object distance ($u$), image distance ($v'$), and radius of curvature ($R_2$) for the convex surface: $\frac{1}{u} + \frac{1}{v'} = \frac{2}{R_2}$

Substituting the values of $u$ and $v'$: $\frac{1}{30} + \frac{1}{10} = \frac{2}{R_2}$

$\frac{1}{6} = \frac{2}{R_2}$

Now, we can solve for $R_2$: $R_2 = \frac{2}{\frac{1}{6}} = 12$ cm

Now, we can substitute the value of $R_2$ back into the equation relating $n$ and $R_2$: $(n-1)\frac{1}{12} = \frac{1}{20}$

Solving for $n$: $n = 1 + \frac{12}{20} = 1 + \frac{3}{5} = \frac{8}{5} = 1.6$

Now, let's analyze the given options:

(A) The refractive index of the lens is 2.5 This is false, as we found the refractive index to be 1.6.

(B) The radius of curvature of the convex surface is $45$ cm This is false, as we found the radius of curvature to be $12$ cm.

(C) The faint image is erect and real This is true. Since the faint image is formed due to reflection from the convex surface, it behaves like a convex mirror. For a convex mirror, the image is always erect and virtual.

(D) The focal length of the lens is 20 cm This is false, as we found the focal length to be $\frac{1}{20}$, which is 0.05 cm.

Final Answer: (C) The faint image is erect and

### A.4.3 Computation Error

Here is an example of a computation error.

Two inductors $L_1$ (inductance 1mH, internal resistance $3\Omega$ ) and $L_2$ (inductance 2mH, internal resistance $4\Omega$ ), and a resistor $R$ (resistance $12\Omega$ ) are all connected in parallel across a 5 V battery. The circuit is switched on at time $t = 0$. What is the ratio of the maximum to the minimum current $(I_{\max}/I_{\min})$ drawn from the battery?

**GPT-4 Response**

Step 1: Analyze the circuit at t=0 At t=0, the inductors behave like short circuits (since they have not had time to build up any magnetic field). Therefore, the current through each inductor is determined by their internal resistances. Let's call the currents through $L_1$, $L_2$, and $R$ as $I_1$, $I_2$, and $I_R$ respectively. Using Ohm's law, we can find these currents:

$$I_1 = \frac{5V}{3\Omega} = \frac{5}{3}A$$

$$I_2 = \frac{5V}{4\Omega} = \frac{5}{4}A$$

$$I_R = \frac{5V}{12\Omega} = \frac{5}{12}A$$

The total current drawn from the battery at t=0 is the sum of these currents:

$$I_{\max} = I_1 + I_2 + I_R = \frac{5}{3} + \frac{5}{4} + \frac{5}{12} = \frac{20}{12}A$$

Wrong calculation, should be $\frac{40}{12}$

Step 2: Analyze the circuit at steady state As time goes to infinity, the inductors will behave like open circuits (since their magnetic fields have reached their maximum and no more current can flow through them). In this case, the only current flowing in the circuit is through the resistor $R$. Using Ohm's law, we can find this current:

$$I_{\min} = \frac{5V}{12\Omega} = \frac{5}{12}A$$

Step 3: Find the ratio of maximum to minimum current Now we can find the ratio of the maximum to the minimum current:

$$\frac{I_{\max}}{I_{\min}} = \frac{\frac{20}{12}A}{\frac{5}{12}A} = \frac{20}{5} = 4$$

So the ratio of the maximum to the minimum current drawn from the battery is 4.

## A.5 GPT-4 verifying its own response

Figures 13, and 14 demonstrate various cases which arise when GPT-4 critiques its own output.

A computer producing factory has only two plants $T_1$ and $T_2$. Plant $T_1$ produces 20% and plant $T_2$ produces 80% of the total computers produced. 7% of computers produced in the factory turn out to be defective. It is known that

$P$ (computer turns out to be defective given that it is produced in plant $T_1$)

$= 10P$ ( computer turns out to be defective given that it is produced in plant $T_2$),

where $P(E)$ denotes the probability of an event $E$. A computer produced in the factory is randomly selected and it does not turn out to be defective. Then the probability that it is produced in plant $T_2$ is

(A) $\frac{36}{73}$
(B) $\frac{47}{79}$
(C) $\frac{78}{93}$
(D) $\frac{75}{83}$

**Answer**

First, let's define the events: - $D$: the computer is defective - $N$: the computer is not defective - $T_1$: the computer is produced in plant $T_1$ - $T_2$: the computer is produced in plant $T_2$

We are given the following probabilities: - $P(T_1) = 0.2$ - $P(T_2) = 0.8$ - $P(D) = 0.07$ - $P(D|T_1) = 10P(D|T_2)$

We want to find the probability $P(T_2|N)$, which is the probability that a computer is produced in plant $T_2$ given that it is not defective.

We can use the Bayes' theorem to find this probability:

$P(T_2|N) = \frac{P(N|T_2)P(T_2)}{P(N)}$

First, let's find $P(N|T_2)$. We know that $P(D|T_2) + P(N|T_2) = 1$, so:

$P(N|T_2) = 1 - P(D|T_2)$

Now, let's find $P(D|T_2)$. We can use the given information that $P(D|T_1) = 10P(D|T_2)$ and the fact that $P(D) = P(D|T_1)P(T_1) + P(D|T_2)P(T_2)$:

$0.07 = 10P(D|T_2)(0.2) + P(D|T_2)(0.8)$

Solving for $P(D|T_2)$, we get:

$P(D|T_2) = \frac{0.07}{2.8} = \frac{1}{40}$

Now we can find $P(N|T_2)$:

$P(N|T_2) = 1 - P(D|T_2) = 1 - \frac{1}{40} = \frac{39}{40}$

Next, let's find $P(N)$. We know that $P(N) + P(D) = 1$, so:

$P(N) = 1 - P(D) = 1 - 0.07 = 0.93$

Finally, we can find $P(T_2|N)$ using the Bayes' theorem:

$P(T_2|N) = \frac{P(N|T_2)P(T_2)}{P(N)} = \frac{\frac{39}{40} \cdot 0.8}{0.93} = \frac{78}{93}$

So the probability that a computer is produced in plant $T_2$ given that it is not defective is $\frac{78}{93}$.

The correct answer is (C) $\frac{78}{93}$.

**Edit**

The given solution is correct, and the answer is (C) $\frac{78}{93}$.

Figure 13: GPT-4 identifying a correct solution as correct.

## A.6 GPT-4 deciding when to attempt

To evaluate if GPT-4 could decide whether to refuse attempt given the negative penalty for being wrong, we prompt it with the following for Single-Correct MCQ questions. `If the answer is wrong, you'll be given -1 marks. If the answer is correct, you'll be given +3 marks. If you're unsure of the answer, you can skip the question, and you'll be given 0 marks.`

`The following is used for Multi-Correct MCQs If any of the options in the final answer is wrong, you'll be given -2 marks. If all the options are correct, you'll be given +4 marks. If some of the options are correct, you'll be given +1 for each correct option. If you're unsure of the answer, you can skip the question, and you'll be given 0 marks.`

**Question**

If $I = \sum_{k=1}^{98} \int_k^{k+1} \frac{k+1}{x(x+1)} dx$, then

[A] $I > \log_e 99$
[B] $I < \log_e 99$
[C] $I < \frac{49}{50}$
[D] $I > \frac{49}{50}$

**Answer**

First, let's evaluate the integral for a single term in the sum:

$\int_k^{k+1} \frac{k+1}{x(x+1)} dx$

We can perform a substitution: let $u = x + 1$, so $x = u - 1$ and $dx = du$. The limits of integration become $u = k + 1$ to $u = k + 2$. The integral becomes:

$\int_{k+1}^{k+2} \frac{k+1}{(u-1)u} du$

Now, we can perform partial fraction decomposition on the integrand:

$\frac{k+1}{(u-1)u} = \frac{A}{u-1} + \frac{B}{u}$

Multiplying both sides by $(u-1)u$ to clear the denominators, we get:

$k + 1 = A(u) + B(u-1)$

To solve for $A$ and $B$, we can use the following system of equations:

1. Set $u = 1$: $k + 1 = B(1-1) \Rightarrow B = 0$ 2. Set $u = 0$: $k + 1 = A(-1) \Rightarrow A = -(k+1)$

So, the integrand becomes:

$\frac{k+1}{(u-1)u} = \frac{-(k+1)}{u-1} + \frac{0}{u} = \frac{-(k+1)}{u-1}$

Now, we can integrate:

$\int_{k+1}^{k+2} \frac{-(k+1)}{u-1} du = -(k+1) \int_{k+1}^{k+2} \frac{1}{u-1} du$

This integral is a simple natural logarithm:

$-(k+1) [\ln |u-1|]_{k+1}^{k+2} = -(k+1)(\ln(k+1) - \ln k)$

Now, we can substitute this back into the original sum:

$I = \sum_{k=1}^{98} (-(k+1)(\ln(k+1) - \ln k))$

Notice that this is a telescoping sum:

$I = -((2\ln 2 - \ln 1) + (3\ln 3 - 2\ln 2) + \cdots + (99\ln 99 - 98\ln 98))$

$I = -(\ln 1 + \ln 2 + \cdots + \ln 98 + \ln 99) = -\ln(1 \cdot 2 \cdot \cdots \cdot 98 \cdot 99)$

Now, let's compare $I$ to $\ln e^{99}$:

$I = -\ln(1 \cdot 2 \cdot \cdots \cdot 98 \cdot 99) < \ln e^{99}$

So, $I < \ln e^{99}$, which means option [B] is correct.

Now, let's compare $I$ to $\frac{49}{50}$:

Since $I = -\ln(1 \cdot 2 \cdot \cdots \cdot 98 \cdot 99)$, we know that $I$ is negative. Therefore, $I < \frac{49}{50}$, which means option [C] is correct.

Thus, the final answer is [B] and [C].

**Gold**: BD

**Edit**

The given solution is incorrect. The error occurs in the step where the integral is evaluated:

$\int_{k+1}^{k+2} \frac{-(k+1)}{u-1} du = -(k+1) [\ln |u-1|]_{k+1}^{k+2} = -(k+1)(\ln(k+1) - \ln k)$

The correct evaluation should be:

$\int_{k+1}^{k+2} \frac{-(k+1)}{u-1} du = -(k+1) [\ln |u-1|]_{k+1}^{k+2} = -(k+1)(\ln(k+2) - \ln(k+1))$

Now, we can substitute this back into the original sum:

$I = \sum_{k=1}^{98} (-(k+1)(\ln(k+2) - \ln(k+1)))$

Notice that this is a telescoping sum:

$I = -((2\ln 3 - \ln 2) + (3\ln 4 - 2\ln 3) + \cdots + (99\ln 100 - 98\ln 99))$

$I = -(\ln 2 + \ln 3 + \cdots + \ln 99 + \ln 100) = -\ln(2 \cdot 3 \cdot \cdots \cdot 99 \cdot 100)$

Now, let's compare $I$ to $\ln e^{99}$:

$I = -\ln(2 \cdot 3 \cdot \cdots \cdot 99 \cdot 100) < \ln e^{99}$

So, $I < \ln e^{99}$, which means option [B] is correct.

Now, let's compare $I$ to $\frac{49}{50}$:

Since $I = -\ln(2 \cdot 3 \cdot \cdots \cdot 99 \cdot 100)$, we know that $I$ is negative. Therefore, $I < \frac{49}{50}$, which means option [C] is correct.

Thus, the final answer is [B] and [C].

Figure 14: GPT-4 identifying a correct step as wrong and making it incorrect

## A.7 Can GPT-4 reason spatially

Figures 15 and 16 show an example problem for which GPT-4's response indicates inability to ground physical concepts spatially.

## A.8 Is JEEBENCH contaminated?

We note that several web-pages on the internet do provide problems and solutions to the JEE exam (for example, here). Therefore, it is possible to find the question and the answer during web-scale training. Unfortunately, this would be true for any dataset that has been created from questions from competitive exams such as (Hendrycks et al., 2021b).

To check if the dataset is indeed contaminated, we perform a careful analysis along the following

A uniform wooden stick of mass 1.6 kg and length $l$ rests in an inclined manner on a smooth, vertical wall of height $h(< l)$ such that a small portion of the stick extends beyond the wall. The reaction force of the wall on the stick is perpendicular to the stick. The stick makes an angle of $30°$ with the wall and the bottom of the stick is on a rough floor. The reaction of the wall on the stick is equal in magnitude to the reaction of the floor on the stick. The ratio $h/l$ and the frictional force $f$ at the bottom of the stick are

($g = 10$ ms$^{-2}$)

(A) $\frac{h}{l} = \frac{\sqrt{3}}{16}, f = \frac{16\sqrt{3}}{3}$ N

(B) $\frac{h}{l} = \frac{3}{16}, f = \frac{16\sqrt{3}}{3}$ N

(C) $\frac{h}{l} = \frac{3\sqrt{3}}{16}, f = \frac{8\sqrt{3}}{3}$ N

(D) $\frac{h}{l} = \frac{3\sqrt{3}}{16}, f = \frac{16\sqrt{3}}{3}$ N

Step 1: Draw a diagram of the situation.

Draw a vertical wall on the left side and a horizontal floor on the bottom. Place the wooden stick inclined at an angle of $30°$ with the wall, with its bottom end on the floor and its top end touching the wall. Label the point where the stick touches the wall as A, the point where the stick touches the floor as B, and the center of mass of the stick as C.

Step 2: Identify the forces acting on the stick.

There are four forces acting on the stick: 1. The gravitational force acting downward at the center of mass C, which is equal to the mass of the stick times the acceleration due to gravity: $mg = 1.6 \times 10 = 16$ N. 2. The normal force exerted by the wall on the stick at point A, acting perpendicular to the stick. 3. The normal force exerted by the floor on the stick at point B, acting perpendicular to the floor. 4. The frictional force exerted by the floor on the stick at point B, acting parallel to the floor and opposite to the direction of the stick's motion.

Step 3: Apply the equilibrium conditions.

Since the stick is in equilibrium, the net force and net torque acting on it must be zero. This gives us two equations:

1. The net force in the horizontal direction is zero: $f = N_A \sin 30°$. 2. The net force in the vertical direction is zero: $N_B = N_A \cos 30° + mg$. 3. The net torque about point B is zero: $N_A \cos 30° \times \frac{l}{2} = mg \times h$.

Step 4: Solve for the unknowns.

We are given that the reaction force of the wall on the stick is equal in magnitude to the reaction of the floor on the stick, i.e., $N_A = N_B$. Using this information, we can solve the above equations for the unknowns $h/l$ and $f$.

From equation 2, we have $N_B = N_A \cos 30° + mg$. Since $N_A = N_B$, we can write this as $N_A = N_A \cos 30° + mg$. Solving for $N_A$, we get $N_A = \frac{mg}{1 - \cos 30°} = \frac{16}{1 - \frac{\sqrt{3}}{2}} = \frac{32}{3}$ N.

Figure 15: Error made by GPT-4 in understanding physical concepts. In this example, correct form of equation 1 should be $f = N_A \cos 30°$. GPT-4 fails to spatially ground the concept of direction in a 2D environment.

**Question**

A farmer $F_1$ has a land in the shape of a triangle with vertices at $P(0,0), Q(1,1)$ and $R(2,0)$. From this land, a neighbouring farmer $F_2$ takes away the region which lies between the side $PQ$ and a curve of the form $y = x^n (n > 1)$. If the area of the region taken away by the farmer $F_2$ is exactly 30% of the area of $\triangle PQR$, then what is the value of $n$?

**Answer**

Step 1: Find the area of triangle PQR.

The area of a triangle with vertices at $(x_1, y_1), (x_2, y_2)$, and $(x_3, y_3)$ can be found using the formula:

$Area = \frac{1}{2}|x_1(y_2 - y_3) + x_2(y_3 - y_1) + x_3(y_1 - y_2)|$

Plugging in the coordinates of P, Q, and R, we get:

$Area_{PQR} = \frac{1}{2}|0(1 - 0) + 1(0 - 0) + 2(0 - 1)| = \frac{1}{2}|-2| = 1$

Step 2: Find the area of the region taken away by farmer $F_2$.

Since the area of the region taken away by farmer $F_2$ is exactly 30% of the area of $\triangle PQR$, we have:

$Area_{F_2} = 0.3 \times Area_{PQR} = 0.3 \times 1 = 0.3$

Step 3: Set up an integral to find the area under the curve $y = x^n$ between P and Q.

The area under the curve $y = x^n$ between P and Q can be found using the integral:

$Area_{curve} = \int_0^1 x^n dx$

Step 4: Calculate the integral and set it equal to the area taken away by farmer $F_2$.

$\int_0^1 x^n dx = \frac{x^{n+1}}{n+1}\Big|_0^1 = \frac{1^{n+1}}{n+1} - \frac{0^{n+1}}{n+1} = \frac{1}{n+1}$

Now, we set this equal to the area taken away by farmer $F_2$:

$\frac{1}{n+1} = 0.3$

Step 5: Solve for n.

To find the value of n, we simply solve the equation:

$n + 1 = \frac{1}{0.3} = \frac{10}{3}$

$n = \frac{10}{3} - 1 = \frac{7}{3}$ ___

Figure 16: Error made by GPT-4 in understanding which curves to take area between. Here it is taking area between the curve and x-axis. However, the question intended between curve and the $x = y$ line. This indicates that GPT-4 might be weak in 2D reasoning from purely text-based prompts.

axes:

1. We search the C4 dataset (Dodge et al., 2021) using 50 randomly sampled questions from Physics and Chemistry(we do not use Math question since they are generally more LaTeX heavy and exact matches would be harder to find). We were unable to find any documents containing sufficiently long substrings from the questions of the dataset.

2. We search the Common Crawl URL Index http://urlsearch.commoncrawl.org/CC-MAIN-2023-23/ for several popular sources which release consolidated full length solutions to exams from the JEE exams. Out of 500+ questions in our dataset, we could find web pages containing the solution of 30 questions [19 chemistry, 11 math] in the latest Common Crawl 2023 index. This is less than 6% of the dataset. We study the effects this has on the final evaluation and observe that removing these questions results in only a

minor decrease in overall scores for most models. This suggests that the contamination doesn't impact the scores greatly. For instance, GPT4+CoT performance goes down from 0.350 to 0.347. The average performance on the 30 questions is 0.392 (this could also be attributed to a much higher proportion of chemistry questions which GPT-4 is better at). It is noteworthy that even though these URLs are present in Common Crawl, that doesn't imply they have been trained on since LLM training doesn't even complete 1 epoch over the pretraining data generally.

3. Since proprietary LLMs do not disclose their exact data recipe, we wish to see if GPT-4 memorized the questions present in the dataset? Note that this is a very challenging problem in itself. Inspired by contemporary methods [6] we take the following heuristic approach: we prompt GPT-4 with a prefix of the question and instruct it to complete the remaining question providing context of the year (for eg, Complete this question from JEE Advanced 2017). In this, we check if the

---

[6]https://github.com/hitz-zentroa/lm-contamination

model is able to generate "precarious" data, that is, data which cannot be predicted from the context. For example, some numerical data provided, or the options of the questions. We use the same 50 questions sampled above and prompt GPT-4. Our evaluation suggested GPT-4 was unable to generate any such responses.

These facts suggest that the extent of contamination, if any, is very low and that is not detrimental to the paper. We will add these analyses in the final paper to clear any aspersions regarding possible contamination.

We additionally want to emphasize that our benchmark is dynamic in the sense that a new set of 40-50 new uncontaminated problems can be added to it annually. This would allow a reliable test of the problem solving abilities of future LLMs as more and more data goes into training them.