# OpenReview forum: "Have LLMs Advanced Enough? A Challenging Problem Solving Benchmark For Large Language Models"
_EMNLP/2023/Conference — EMNLP 2023 Main_

### Official Review · Reviewer_8fFX · 2023-07-27

**Soundness:** 4

**Excitement:**

4: Strong: This paper deepens the understanding of some phenomenon or lowers the barriers to an existing research direction.

**Paper Topic And Main Contributions:**

The paper's primary contribution is a dataset of IIT JEE-Advanced questions, many of which appear to require sophisticated step-by-step reasoning and a fairly broad and thorough analysis of SoTA models on this dataset.

**Questions For The Authors:**

See reasons to reject

**Reasons To Accept:**

We're running into the limits of MMLU and other broad-coverage datasets evaluating model reasoning. We desperately need more good, uncontaminated datasets to evaluate LLM reasoning. The methodology appears solid; the analysis is very interesting (e.g., the discussion around self-critique and contamination); the writing is good.

**Reasons To Reject:**

I am a bit disappointed that problems with diagrams were eliminated. It would have been nice to have these separately included, especially with the emergence of more and more multimodal models. It's also a bit unclear to me what the planned release mechanism is for this dataset. Is it going to be completely publicly released? Is the test answer set going to be held out like in SuperGLUE?

**Reproducibility:**

4: Could mostly reproduce the results, but there may be some variation because of sample variance or minor variations in their interpretation of the protocol or method.

**Reviewer Confidence:**

3: Pretty sure, but there's a chance I missed something. Although I have a good feel for this area in general, I did not carefully check the paper's details, e.g., the math, experimental design, or novelty.

---

> ### Author Rebuttal · Authors · 2023-08-29
>
> We thank the reviewer for their positive comments. We are glad that they found our methodology robust, our analysis interesting.
>
> The following points provide responses to the reviewer's questions:
> - **Removal of questions with images**: We focused on text-only questions in this paper with the motivation that the reasoning ability of text-only models has improved very significantly compared to multimodal vision-language models. That said, we emphasize that collecting image-based questions is indeed feasible and we keep this for future work.
>
> - **Public release of the dataset**: Yes, the dataset will be completely publicly released. The data is purely meant as a test-set. There is no train/val split for the data. What we meant by the planned release mechanism was that the dataset could be updated every year with new questions pouring in from the examination.

---

### Official Review · Reviewer_Kktx · 2023-08-03

**Soundness:** 4

**Excitement:**

4: Strong: This paper deepens the understanding of some phenomenon or lowers the barriers to an existing research direction.

**Missing References:**

- Discussion and references of previous works on calibration and risk assessment are warranted. For example, [1] thoroughly addresses calibration and self-knowledge assessment and confirms the high calibration ability of LLMs. Discussion in Line 539 could also be expanded to address findings from recent work.

[1] https://arxiv.org/pdf/2207.05221.pdf

**Paper Topic And Main Contributions:**

The authors introduce a benchmark based on the IIT JEE-Advanced exam, which provides a sufficient challenge for current LLMs, including GPT-4 and PaLM-2. The authors provide a comprehensive evaluation of the latest open- and closed-source LLMs, using greedy decoding, as well as popular state-of-the-art techniques such as self-consistency and self-critiquing. Here, the authors provide a well-organized and extensive qualitative study on errors in initial solutions *and* critiques, using 100 random samples. In addition to accuracy and calibration performance, the authors probe the LLMs' ability to assess their own abilities and risk, through evaluation under a negative marking scheme (i.e., negative scores for wrong answers, with option to abstain). The authors consider a prompt-based solution and confidence thresholding (based on the frequency of answers from repeated inferences) and find that the latter is effective, and the former is not.

In summary, the paper introduces a challenging benchmark that looks to be useful for evaluating the problem solving and self-knowledge-assessing abilities of current and future language models. The paper provides an extensive analysis of current models based on the proposed benchmark and various popular inference methods.

**Questions For The Authors:**

- E. Will the dataset *and* code for experiments be made available to the public? There is no link.
- F. I'm concerned about the copyrights of the data, as it is not addressed. Do you own, or have obtained permission to publish, the data?

**Reasons To Accept:**

- The benchmark evaluates many high-level problem-solving abilities (domain-specific concepts, mathematical grounding and computing) and is sufficiently challenging to effectively differentiate current language models and leaves much room for improvement by future LLMs. This is very important for assessing and guiding the development of LLMs.
- The authors adequately discuss contamination through the inclusion of recent data and separate model evaluation.
- Extensive evaluation of current models from various aspects (performance, calibration, assessment of self-knowledge), covering a wide range of popular techniques (chain-of-thought, self-consistency, self-critiquing).
- Extensive qualitative evaluation of error cases regarding not only the initial solutions but also the self-critiques. The taxonomy of errors provided by the authors is notably clear and insightful, which makes the qualitative analysis informative and easy to read.
- The paper is very well-written.

**Reasons To Reject:**

- A. Few-shot prompted evaluations are not provided. This is important to validate the utility of this benchmark, i.e., whether it is challenging enough. Though the authors cite cost and time as a reason for this omission in Line 270, however, the authors include experiments on self-consistency with 8 repetitions, which I expect would cost more than 8-shot prompting. Also, I do not believe that inference on 500 test samples would take too long, even with the longer length. Results on few-shot prompting (even with very few e.g. 1, 2) would improve the validity of the challenge and provide insights on the effect of few-shot prompting. Note that current results do already *suggest* sufficient difficulty.
- B. There is limited analysis of the difference in performance *between* different models. Some analysis, or at least a discussion on the reasons behind the performance trends may be insightful.

### Minor

- C. It would be interesting to see the effect of instruction tuning on model performance. Despite mentioning that the results "demonstrate the "efficacy of ... instruction fine-tuning and RLHF" (Line 511), there is no comparison of vanilla and instruction-tuned models. Note, "GPT-3" used in the paper (text-davinci-003) is another *instruction-tuned* variant of GPT-3.
- D. Multi-choice questions leave a significant chance for the models to simply guess the right answer (I assume that is why 28% of correct answers come from incorrect reasoning, Line 343). It would help readers understand the true performance of models if you could include baseline random-chance performance, or normalize the scores based on random-chance probability p via (accuracy - p) / (1 - p), following other papers such as BIG-Bench.

**Reproducibility:**

4: Could mostly reproduce the results, but there may be some variation because of sample variance or minor variations in their interpretation of the protocol or method.

**Reviewer Confidence:**

4: Quite sure. I tried to check the important points carefully. It's unlikely, though conceivable, that I missed something that should affect my ratings.

**Typos Grammar Style And Presentation Improvements:**

- Figures have low image resolution. They are slightly blurry on print and high-res displays. Authors could consider non-raster graphics.

### Complements

- I like that you put GPT-4 responses in the text of the Appendix rather than a figure, as many papers do. I think it's more readable!

---

> ### Author Rebuttal · Authors · 2023-08-29
>
> We thank the reviewer for the generous and detailed review. We are thrilled by the fact that they found our work important for assessing LLMs, our discussions thorough and our evaluation extensive. We apologize for the low fidelity of the images. We will work on improving them in the final version of the paper.
>
> The following points provide responses to the reviewer’s questions.
>
> **Few shot prompting:**
>
> We perform the following experiment for few shot prompting:
>
> For each (subject, question_type) pair, we collect an example question from JEE Advanced 2014 and write their solutions. These are then used for 1-shot prompting. Overall, 1-Shot CoT achieves a score of 0.296 as opposed to zero-shot CoT(Just adding "Let's think steo by step") at 0.350 and vanilla GPT-4 at 0.308. Our hypothesis is that Few-shot prompting is not very helpful in these questions because conceptual errors are hard to be improved upon using few-shot examples. Additionally, many novel reasoning paths might not be covered in the few-shot examples. Thus, our dataset acts as an interesting testbed for advanced approaches in few-shot prompting. Similar results where results are better with zero-shot CoT as compared to few-shot CoT have been found in [SciBench](https://arxiv.org/abs/2307.10635)(See Table 3). We intend to add this discussion to our paper.
>
> **Difference between models**
>
> We would like to draw the reviewer’s attention towards Table 2 where we provide numerical comparison of performance on JEEBench across various open-source and proprietary LLMs. We qualitatively compare these in Section 4.2 under the subsection “General Trends''.  Our observation is that open-source models perform as good as random and are, in general, lagging behind proprietary models. Performance on JEEBench increases consistently with newer versions of the GPT model. GPT-3 exhibits near random performance, but GPT-3.5 and GPT-4 perform significantly better, GPT-4 being far superior. It is evident that the performance boost is the highest for Chemistry, followed by Physics, and lastly Maths. This is probably because the complexity of reasoning is highest in Mathematics questions and least in Chemistry in JEEBench.
>
> **Effect of instruction tuning**
>
> Instruction-tuning can be done in many ways, one of them being PPO using a reward model as shown in the [InstructGPT](https://arxiv.org/abs/2203.02155) paper. For this aspect of LLMs, we perform inference using davinci-002, which is the base model for GPT-3. text-davinci-003 has been derived from davinci-002 using PPO on a reward model trained from human-feedback as documented by OpenAI [here](https://platform.openai.com/docs/model-index-for-researchers/instructgpt-models). The responses generated show that davinci-002 almost never reach the answer when prompted with the instruction (such as, "In this problem, only one option will be correct. Give a detailed solution and end the solution with the final answer"). Therefore, its performance is extremely poor (near zero) as compared to text-davinci-003. We will add this analysis to our paper.
>
> **Random performance**
>
> We would like to point out that we have already added the random performance of the models in Table 2. We would include the normalized metric: (accuracy-p)/(1-p) in the final version of the paper.
>
> **Will code and dataset, both be released?**
>
> Yes. We will add the link to the code, dataset and evaluation scripts in the final version of the paper.
>
> **Permissions to use this data**
>
> The JEE Examination is a national exam which is conducted every year, and the exam papers are distributed for the general public (https://www.jeeadv.ac.in/archive.html) and has been used by various publishers and institutes for commercial purposes already. We also have clarifications from official sources that we are allowed to use this data since it is publicly available.
>
> **Missing references for calibration studies**
>
> We apologize for not providing adequate references for our calibration study. We would add these in the final version of the paper.

---

### Official Review · Reviewer_Yhxr · 2023-08-07

**Typos Grammar Style And Presentation Improvements:** NA
**Soundness:** 4

**Excitement:**

4: Strong: This paper deepens the understanding of some phenomenon or lowers the barriers to an existing research direction.

**Missing References:**

There are a few concurrent works when this paper is submitted, including:
[1] C-Eval: A Multi-Level Multi-Discipline Chinese Evaluation Suite for Foundation Models
[2] SciBench: Evaluating College-Level Scientific Problem-Solving Abilities of Large Language Models
[3] Exploring the MIT Mathematics and EECS Curriculum using Large Language Models

and one older work:
[4] Galactica: A Large Language Model for Science

These works also provide some scientific questions. I think it's better to highlight some difference of this work compared with them.

**Paper Topic And Main Contributions:**

This paper proposes a new LLM benchmark called JEEBench, which comprises of pre-engineering mathematics, physics, and chemistry problems derived from the highly competitive IIT JEE-Advanced exam, testing LLM's long-horizon reasoning and in-depth domain knowledge. The paper also introduces an innovative method for effective response selection by applying confidence-thresholding over self-consistency. The authors have identified and detailed the common failure modes of GPT-4, such as errors in algebraic manipulation, difficulty in translating abstract concepts into mathematical equations, and failing to retrieve relevant domain-specific concepts.

**Questions For The Authors:**

1. Could authors provide some justification that the exam & answer they used are not easily accessed in LLM pre-training? (e.g., answer is located in different places from question)
2. The authors mention the calculation capability of LLM. Could the authors try adding some calculation tools (like python / wolfram alpha) to LLM and see its performance?
3. The examples are blurred. It would be better to provide clear examples for such a dataset paper.

**Reasons To Accept:**

In the era of LLM, a comprehensive benchmark is very necessary for NLP community. This paper focus on IIT JEE-Advanced exam, which is both hard and broad enough to test different capabilities of LLM.

**Reasons To Reject:**

Overall I think it's a good paper, with nice data and thorough analysis.

Only concern is whether the dataset might be appeared in training corpus of LLM, which I suggest authors provide a few justification.

**Reproducibility:**

4: Could mostly reproduce the results, but there may be some variation because of sample variance or minor variations in their interpretation of the protocol or method.

**Reviewer Confidence:**

5: Positive that my evaluation is correct. I read the paper very carefully and I am very familiar with related work.

---

> ### Author Rebuttal · Authors · 2023-08-29
>
> We thank the reviewer for their insightful comments and suggestions. We appreciate the fact that the reviewer finds our dataset nice and analysis thorough.
>
> The following points provide clarifications to concerns raised by the reviewer.
>
> **The dataset could be present in LLM pre-training**
>
> Firstly, please note that the dataset requires significant preprocessing for converting into a LaTeX format. This occasionally requires human intervention in some cases due to errors made by automatic tools.
>
> Further, we wish to draw the reviewer’s attention to Section 4.7 of our manuscript where we mention that the performance of GPT-4 on JEE’23 (conducted in July 2023, so definitely not in the pre-training data) is sufficiently close to its average performance on the past years indicating minimal (if any) memorization.
>
> We note that several web-pages on the internet do provide problems and solutions to the JEE exam. Here is an example: https://www.toppr.com/ask/question/highly-excited-states-for-hydrogenlike-atoms-also-called-ryberg-states/. Therefore, it is possible to find the question and the answer during web-scale training. Unfortunately, this would be true for any dataset that has been created from questions from competitive exams (for example MATH, AGIEval).
>
> To check if the dataset is indeed contaminated, we perform a careful analysis along the following axes:
>
> **Are the solutions present in popular open-source pretraining datasets?**
>
> 1. We search the C4 dataset (https://c4-search.apps.allenai.org/) using 50 randomly sampled questions from Physics and Chemistry(we do not use Math question since they are generally more LaTeX heavy and exact matches would be harder to find). We were *unable to find any documents* containing sufficiently long substrings from the questions of the dataset.
>
> 2. We search the Common Crawl URL Index (http://urlsearch.commoncrawl.org/) for 6 popular sources which release consolidated full length solutions to exams from the JEE exams: Aakash, Byju’s, FIITJEE, Resonance, Vedantu, and Vidyamandir. Out of 500+ questions in our dataset, we could find web pages containing the solution of 30 questions [19 chemistry, 11 math] in the latest Common Crawl 2023 index. This is less than 6% of the dataset.
> We study the effects this has on the final evaluation and observe that removing these questions results in only a minor decrease in overall scores for most models. This suggests that the contamination doesn’t impact the scores greatly. For instance, GPT4+CoT performance goes down from 0.350 to 0.347. The average performance on the 30 questions is 0.392(this could also be attributed to a much higher proportion of chemistry questions which GPT-4 is better at). Even though there isn't much difference in the performance on (possibly) contaminated and (possibly) non-contaminated questions, we will remove them from the dataset.
>
> It is noteworthy that even though these URLs are present in Common Crawl, that doesn’t imply they have been trained on since LLM training doesn’t even complete 1 epoch over the pretraining data generally.
>
>
> Since proprietary LLMs do not disclose their exact data recipe, we wish to see if GPT-4 memorized the questions present in the dataset? Note that this is a very challenging problem in itself. Inspired by contemporary methods(https://hitz-zentroa.github.io/lm-contamination/blog/) we take the following heuristic approach: we prompt GPT-4 with a prefix of the question and instruct it to complete the remaining question providing context of the year (for eg, Complete this question from JEE Advanced 2017). In this, we check if the model is able to generate “precarious” data, that is, data which cannot be predicted from the context. For example, some numerical data provided, or the options of the questions. We use the same 50 questions sampled above and prompt GPT-4. Our evaluation suggested GPT-4 was unable to generate any such responses.
>
> These facts suggest that the extent of contamination, if any, is very low and that is not detrimental to the paper. We will add these analyses in the final paper to clear any aspersions regarding possible contamination.
>
> We additionally want to emphasize that our benchmark is dynamic in the sense that a new set of ~40-50 new uncontaminated problems can be added to it annually. This would allow a reliable test of the problem solving abilities of future LLMs as more and more data goes into training them.
>
> **Adding calculation tools**
>
> The JEE exam only allows access to a basic calculator with 4 primitive arithmetic operation(+, -, /, *). Using plugins such as Wolfram or python would not make it a fair comparison(for example, directly asking Wolfram to integrate a function instead of doing it from first principles). Instead, to ensure a level-playing field, we use the newly introduced [function-calling API](https://openai.com/blog/function-calling-and-other-api-updates) by GPT-4 to implement standard arithmetic operators.
>
> Note that function-calling is only allowed with gpt-4-0613 whereas our initial results were with gpt-4-0314. We first evaluate gpt-4-0613 with zero-shot CoT. Surprisingly, results suggest that gpt-4-0613 is weaker than the previous model with a CoT performance of 0.303 as compared to 0.350 with gpt-4-0314.
>
> Using a calculator API reduces performance even more to 0.274 from 0.303. We observe that tool usage is not very robust for GPT-4, where it sometimes hallucinates invalid function arguments and sometimes even invalid function names! Also, GPT-4 is quite accurate at arithmetic for small digit operations. Computation errors are mostly at a “symbolic” level than at an arithmetic level which is probably why a black-box calculator might not be very beneficial. We hypothesize that the way gpt-4 integrated function-calling might be degrading the prompt which leads to worse performance.
>
>
> **Blurred Examples**
>
> We apologize to the reviewers for the blurred images. We would replace them with higher fidelity images in the final version of the paper.
>
>
> **Missing references**
>
> We thank the reviewers for suggesting more references in the manuscript. We find C-Eval and SciBench to be very relevant concurrent works. C-Eval is a dataset built on problems in Chinese, derived from Chinese college and professional exams. SciBench is made along similar lines borrowing data from famous books on Physics, Chemistry and Mathematics. The paper on Exploring the MIT EECS curriculum attempts similar analysis on questions taken from MIT Exams. We will cite all these papers.
>
> Galactica is a language model which has been fine-tuned on a lot of technical data such as papers, knowledge bases and reference materials and shows significant improvements over GPT-3 and PaLM on relevant datasets such as MMLU and Math. However, the model is not publicly available, which is why we can't run inference on it. Even then, we find it is very relevant and will cite it.

---

### Meta-Review · Area_Chair_q2t4 · 2023-09-23

**Recommendation:** 5

**Metareview:**

This paper proposes a new LLM benchmark called JEEBench, which comprises of pre-engineering mathematics, physics, and chemistry problems derived from the highly competitive IIT JEE-Advanced exam, testing LLM's reasoning and in-depth domain knowledge. The paper also introduces an innovative method for effective response selection by applying confidence-thresholding over self-consistency. The authors have identified and detailed the common failure modes of GPT-4, such as errors in algebraic manipulation, difficulty in translating abstract concepts into mathematical equations, and failure to retrieve relevant domain-specific concepts.

In the era of LLM, a comprehensive benchmark is very necessary for NLP community. This paper focuses on IIT JEE-Advanced exam, which is both hard and broad enough to test different capabilities of LLM.

All reviewers found the paper strong and exciting.

---

### Decision · Program_Chairs · 2023-10-07

**Decision:**

Accept-Main

**Comment:**

This paper proposes a new LLM benchmark called JEEBench, which comprises of pre-engineering mathematics, physics, and chemistry problems derived from the highly competitive IIT JEE-Advanced exam, testing LLM's reasoning and in-depth domain knowledge. The paper also introduces an innovative method for effective response selection by applying confidence-thresholding over self-consistency. The authors have identified and detailed the common failure modes of GPT-4, such as errors in algebraic manipulation, difficulty in translating abstract concepts into mathematical equations, and failure to retrieve relevant domain-specific concepts.

In the era of LLM, a comprehensive benchmark is very necessary for NLP community. This paper focuses on IIT JEE-Advanced exam, which is both hard and broad enough to test different capabilities of LLM.

All reviewers found the paper strong and exciting.